# The Involvement of Cell-Type-Specific Glycans in *Hydra* Temporary Adhesion Revealed by a Lectin Screen

**DOI:** 10.3390/biomimetics7040166

**Published:** 2022-10-15

**Authors:** Sofia Seabra, Theresa Zenleser, Alexandra L. Grosbusch, Bert Hobmayer, Birgit Lengerer

**Affiliations:** 1Institute Superior Técnico, University of Lisboa, Av. Rovisco Pais, 1049-001 Lisboa, Portugal; 2Institute of Zoology and Center of Molecular Biosciences Innsbruck, University of Innsbruck, Technikerstr. 25, 6020 Innsbruck, Austria

**Keywords:** underwater bioadhesion, reversible attachment, adhesives, glue, carbohydrates

## Abstract

*Hydra* is a freshwater solitary polyp, capable of temporary adhesion to underwater surfaces. The reversible attachment is based on an adhesive material that is secreted from its basal disc cells and left behind on the substrate as a footprint. Despite *Hydra* constituting a standard model system in stem cell biology and tissue regeneration, few studies have addressed its bioadhesion. This project aimed to characterize the glycan composition of the *Hydra* adhesive, using a set of 23 commercially available lectins to label *Hydra* cells and footprints. The results indicated the presence of N-acetylglucosamine, N-acetylgalactosamine, fucose, and mannose in the adhesive material. The labeling revealed a meshwork-like substructure in the footprints, implying that the adhesive is mainly formed by fibers. Furthermore, lectins might serve as a marker for *Hydra* cells and structures, e.g., many labeled as glycan-rich nematocytes. Additionally, some unexpected patterns were uncovered, such as structures associated with radial muscle fibers and endodermal gland cells in the hypostome of developing buds.

## 1. Introduction

The ability of organisms to attach to surfaces is entitled biological adhesion and is present in various organisms. Bioadhesion processes are diverse and complex and play a crucial role in organism survival and basic functions [1]. In glue-based adhesive systems, the attachment is mediated by the secretion of an adhesive material and can either be temporary or permanent [2]. The biochemical composition of the secreted adhesive material varies among organisms and is difficult to characterize [3]. It is generally stated that in temporarily adhering animals, such as *Hydra*, the glue is mainly constituted of proteins and carbohydrates [2,4]. Studies on aquatic temporary adhesives predominately focus on the identification of proteins, but, especially in temporary adhesion, carbohydrates are abundant in the secreted material [2]. Adhesion-related glycans have mostly been detected through histological stains such as Alcain blue and lectin-binding assays [2]. Using lectin-based methods, glycans have consistently been detected in the adhesive of non-permanently adhering animals such as sea urchins [5,6], sea stars [7,8], flatworms [9,10,11], and limpets [12]. Moreover, aquatic adhesive proteins are often highly glycosylated [5,6,7,13,14]. This post-translational modification significantly changes proteins characteristics and has to be taken into account in any biomimetic approach. 

The cnidarian *Hydra* is a solitary polyp, inhabiting shallow freshwater bodies. It attaches itself to underwater surfaces through the secretion of an adhesive material and can repeatedly voluntarily detach and reattach [15]. *Hydra* is a classic and simple model system for pattern formation, regeneration, and stem cell biology research [16,17,18,19,20]. Structurally, *Hydra* has a single apical-basal axis with radial symmetry, and two layers of epithelial cells (the endoderm and the ectoderm) separated by an extracellular matrix (the mesoglea) (Figure 1). The *Hydra* body is composed by the head, the gastric region, and the foot, where the animals attaches itself with its basal disc (Figure 1A,B). The basal disc cells produce and secrete the adhesive material, and four morphological distinct secretory granule types (HSGI to IV) have been described. The cells are characterized by an irregular rectangular-like shape, water vacuoles, and numerous secretory granules accumulating at the aboral end, the area of attachment [15]. In contrast to animals with a duo-gland adhesive system, *Hydra* lacks dedicated de-adhesive gland cells that secrete a substance to weaken the bond between the animals and the substrate [4]. In *Hydra*, the detachment process likely occurs due to muscle contractions [15]. It was proposed that the individual basal disc cells retract from the surface, with the movement starting at the outer rim of the basal disc and moving towards the center. Upon detachment, an underwater transparent footprint, composed by the secreted adhesive material, is left on the substrate. The footprint is formed by the secretion and blending of the contents of the adhesive granules [15]. Expression analysis in combination with mass spectrometry of the secreted footprints revealed 21 footprint-specific proteins [21]. These proteins presumably ensure adhesion and cohesion, and contain domains that mediate protein–protein and protein–carbohydrate interactions. Remarkably, eight of these proteins are annotated with glycan-binding domains, such as galactose and chitin binding domains [21]. Periodic acid Schiff (PAS) staining revealed the presence of glycans within some of the basal disc secretory granules, but the glycan composition of the *Hydra* adhesive is unkown [15,22]. 

Here, we characterize the glycan composition of the *Hydra* adhesive material, using a set of 23 commercially available lectins. We applied the lectins to label *Hydra* tissue, including whole-mount animals and macerated basal disc cells, and the secreted footprints. Overall, eight lectins detected the footprints left behind on the substrate, indicating the presence of N-acetylglucosamine, N-acetylgalactosamine, fucose, and mannose in the adhesive material. The secreted adhesive appeared fibrillar and formed a dense meshwork, with an accumulation of material at the cell borders of the formerly attached basal disc cells. Furthermore, our results indicated a high abundance of glycans within mature nematocytes and revealed some unexpected pattern, for example glycans associated with radial muscle fibers or within the hypostome of developing buds. 

## 2. Methods

### 2.1. Hydra Culture

All experiments were carried out with individuals of *Hydra magnipapillata* strain 105, which were bred and kept in mass cultures at the Institute of Zoology, University of Innsbruck. *Hydra* cultures were kept in growth chambers at 18 °C in *Hydra* culture medium and fed five times per week with *Artemia nauplii*. Before any experiment, the animals were starved for 24 h. 

### 2.2. Whole-Mount Lectin Labeling

For the 23 used lectins, the full names, abbreviations, and sugar specificities are listed in Appendix A. Whole-mount animals were relaxed in 2% urethane in culture medium for 3–5 min. They were subsequently fixed using three different conditions: 4% paraformaldehyde (PFA) in phosphate buffered saline (PBS) overnight at 4 °C, 4% PFA in PBS for 1 h at room temperature (RT) and Lavdowsky fixative (ethanol:formamid:acetic acid:distilled water—50:10:4:40) for 4 h at RT. Samples were washed several times in Tris-buffered saline (TBS, pH 8.0) supplemented with 5 mM CaCl_2_ and 0.1% Triton X (TBS-T). Unspecific background staining was blocked by pre-incubation in TBS-T containing 3% (w/v) bovine serum albumin (BSA) for 2 h at RT. Biotinylated lectins (Vector) were diluted in BSA-T to a final concentration of 10 (for WGA) or 25 µg/mL (all other lectins) and applied to the samples for 2 h at RT. After three washes of 10 min each in TBS-T (with 0.05% Triton X), the animals were incubated in Dylight488-conjugated-streptavidin (Vector), Phalloidin-Atto 565, and DAPI diluted (1:500, 1:1000, and 1:10,000, respectively) in BSA-T for 1 h at RT. After three washes in TBS-T (0.05% Triton X), the samples were mounted in Vectashield. During the waiting periods, the samples were placed on a shaker (40 rot/min). Control reactions were performed by substituting the lectins with TBS-T-BSA. The samples were analyzed with a Leica DM5000 microscope or with a Leica SP5 II confocal scanning microscope. As the intensity of the labeling varied among different lectins (see Table 1), the images of the most strongly stained specimens (+++) and of weakly stained (+) specimens had to be taken at different exposure times to sufficiently visualize them without over- or underexposure. The negative control images were taken with the same, longer exposure time as the weakly stained specimen. With the Leica SP5 II confocal scanning microscope, z-stacks were acquired and maximum z-projected.

### 2.3. Footprint Lectin Labeling

Footprints were collected by placing animals on microscope glass slides, submerged in *Hydra* medium and allowing them to attach overnight. The next day, the animals were gently detached using a glass pipet and the slides were washed with distilled water. The slides were then fixed in 4% PFA in PBS for 1 h at RT. The labeling was performed as for the whole-mount lectin labeling, but without adding phalloidin and DAPI. Footprint double stainings were performed using biotinylated lectins (Vector) (25 µg/mL) and Starlight conjugated Streptavidin (Vector) (1:500 diluted) and fluorescein conjugated WGA (Vector) (10 µg/mL). 

### 2.4. Hydra and Corresponding Footprint Labeling after Voluntary and Forced Detachment

*Hydras* were placed on microscope glass slides, submerged in *Hydra* medium and allowed to attach under observation. Whenever a *Hydra* detached voluntarily, the animal and the glass slides were instantly fixed in PFA for 1 h at RT. Attachment times for voluntarily detached animals ranged from 2 to 33 min (*n* = 15). Additionally, *Hydras* were forcibly detached by the investigator after 35 to 95 min attachment time (*n* = 15) and animals and glass slides were fixed the same way. Samples were washed several times in TBS-T and blocked in BSA-T for 1 h at RT. The samples were then incubated in fluorescein conjugated WGA (Vector) (10 µg/mL) in BSA-T for 2 h at RT, washed several times and mounted in Vectashield. The labeling was analyzed with a Leica DM5000 microscope. 

### 2.5. Single Basal Disc Cells Lectin Labeling

Fifteen budless polyps were cut at 20% of the body column, separating the foot from the anterior part. The basal disc cells were obtained by maceration of a single *Hydra* foot, by adding 1 drop of maceration solution (acetic acid:glycerol:distilled water—1:1:14) per animal [23]. Upon 1 h of incubation at RT, cell separation was driven by mechanical forces, by gently tapping on the tube and gently pipetting up and down. For fixation, 1 drop of 8% formaldehyde was added per drop of maceration solution. A total of 50 µL of the resulting solution was added to a pre-treated slide (coated with gelatin), spread into a rectangulare shape and left to dry for 1 h at RT. After this, lectin staining was performed following the protocol used for the footprints (starting at the first washing step). The labeling was analyzed with a Leica SP5 II confocal scanning microscope.

## 3. Results

### 3.1. Lectin Labeling of Hydra Tissues and Footprint Secretions

We performed lectin labeling of whole-mount animals and macerated basal disc cells, and footprints using 23 commercially available lectins. As the fixative and fixation duration can influence the outcome of the labeling, we performed the whole-mount labeling under three standard conditions: animals fixed with PFA over night at 4 °C, fixed with PFA for one hour at room temperature, and fixed with Lavdowsky for four hours at room temperature. For maceration experiments, the best results (regarding cell morphology and staining intensity) were obtained with a fixation with formaldehyde. Footprints were labeled without fixation and after one hour of PFA fixation at room temperature, without any apparent difference in the results (Appendix A). All labeling results are summarized in Table 1, indicating the intensity of the staining of the different cells and structures. Details on the sugar moieties recognized by the lectins are listed in Table 1. 

**Table 1 biomimetics-07-00166-t001:** Overview of lectin labeling in *Hydra* whole-mounts, basal disc cells, and footprints. +++, strong labeling, ++ intermediate labeling, + weak labeling, X not performed. a—patchy staining, associated with muscle fibers; b—cell membranes; c—subcellular structures; d—gland cells in the hypostome of developing buds; e—base of cnidocil; f—erupted nematocytes’ tubules; g—dot-like; h—secretory granules; i—variant intensity among individual footprints.

			Gastric Region			Tentacles				Foot		
Lectin	Acronym	Fixative	Endoderm	Overall Ectoderm Surface	Nematocytes, Nematoblasts or Vacuoles	Tentacle Surface	Nematocytes			Basal Disc Surface	Basal Disc Cells	Footprints
							Capsules	Opercolum	Cnidocil			
Wheat germ agglutinin	WGA	o.n PFA		**+++**		**+++ g**	**++**	**++**	**++**	**++**	**++ h**	**+++**
		1 h PFA		**+++**		**+++ g**	**++**	**++**	**++**	**++**		
		4 h Lavdwosky			**+++**	**+++ g**	**++**	**++**	**++**	**++**		
Succinylated wheat germ agglutinin	sWGA	o.n PFA		**+++**		**++ g**	**+++**	**++**		**++**	**++ h**	**+++**
		1 h PFA		**+++**		**++ g**	**+++**	**++**		**++**		
		4 h Lavdwosky			**+++**	**++ g**	**+++**	**++**		**++**		
*Datura Stramonium* lectin	DSL	o.n PFA				**++ g**	**++**	**++**			**++**	**+ i**
		1 h PFA		**+**		**++ g**	**++**	**++**				
		4 h Lavdwosky		**++ g**		**++ g**	**++**	**++**				
*Lycopersicon esculentum* (tomato) lectin	LEL	o.n PFA					**+++**				**+**	**+++**
		1 h PFA		**+++ g**		**+++ g**	**+++**			**++**		
		4 h Lavdwosky	**+++ b**				**+++**					
Soybean agglutinin	SBA	o.n PFA		**++ g**		**++ g**	**+++**	**+++**			**+++**	**+++**
		1 h PFA		**++ g**		**++ g**	**+++**	**+++**				
		4 h Lavdwosky		**++ g**		**++ g**	**+++**	**+++**				
*Ricinus communis* agglutinin	RCA	o.n PFA		**+**						**+**	**+++ h**	**+ i**
		1 h PFA		**+**						**+**		
		4 h Lavdwosky		**+**						**++**		
*Ulex europaeus agglutinin I*	UEA I	o.n PFA		**+++**							**+++ h**	**+ i**
		1 h PFA		**+++**								
		4 h Lavdwosky		**+++**			**++ f**		**++**			
Concanavaline A	Con A	o.n PFA		**+**		**+++**	**+**	**++**	**+++**	**++**	**+++**	**++**
		1 h PFA		**+**		**+++**	**+**	**++**	**+++**	**++**		
		4 h Lavdwosky		**+**		**+++**	**+**	**++**	**+++**	**++**		
*Erythrina cristagalli* lectin	ECL	o.n PFA				**+++ g**					**x**	
		1 h PFA		**++ g**		**++ g**				**+++ g**		
		4 h Lavdwosky		**++ g**	**+++**	**++ g**				**+++ g**		
*Pisum sativum* agglutinin	PSA	o.n PFA		**+**		**++**			**+++**	**++**	**x**	
		1 h PFA		**++**		**++**			**++**			
		4 h Lavdwosky			**+++ f**	**++**						
*Griffonia* (Bandeiraea) *simplicifolia* lectin I	GSL I	o.n PFA		**+**			**+++**	**+++**		**+++**	**x**	
		1 h PFA		**+**			**+++**	**+++**		**+++**		
		4 h Lavdwosky			**+**		**++**	**+++**		**+++**		
*Dolichos bilforus* agglutinin	DBA	o.n PFA		**++**			**+++**	**+++**			**x**	
		1 h PFA			**++**		**+++**	**+++**		**++**		
		4 h Lavdwosky			**++**		**+++**	**+++**		**+**		
*Phaseolus vulgaris* erythro agglutinin	PHA-E	o.n PFA	**+++ a**	**++**							**x**	
		1 h PFA	**+++ a**	**++**								
		4 h Lavdwosky	**+++ b**	**+++**								
*Phaseolus vulgaris* leuco agglutinin	PHA-L	o.n PFA	**+++ a,b**	**++**							**x**	
		1 h PFA	**+++ a**	**++**								
		4 h Lavdwosky	**+++ b**	**+++ b**								
*Griffonia* (*Bandeiraea*) *simplicifolia* lectin II	GSL II	o.n PFA			**++ c**						**x**	
		1 h PFA			**++ c**							
		4 h Lavdwosky	**++ d**		**++**							
*Lens culinaris* agglutinin	LCA	o.n PFA							**+++ e**		**x**	
		1 h PFA							**+++ e**			
		4 h Lavdwosky			**++**	**++**	**++ f**					
*Vicia villosa* agglutinin	VVL	o.n PFA		**+**		**+++ g**	**++**	**++**	**+++**		**x**	
		1 h PFA		**+**		**+++ g**	**++**	**++**	**+++**			
		4 h Lavdwosky		**++**		**+++ g**	**++**	**++**				

Out of the 23 tested lectins, 17 led to a distinct labeling in whole-mount animals (Table 1) and six (Elderberry bark lectin, Jacalin, *Maackia amurensis* lectin II, Peanut agglutinin, *Sonaum tuberosum* lectin, *Sophora Japonica* agglutinin) did not react with any *Hydra* tissue (Appendix A). However, *Sonaum tuberosum* lectin (STL) led to blurry staining surrounding erupted nematocytes’ threads, potentially reacting with the capsules’ contents (Appendix A). Due to our focus on *Hydra* adhesive secretions, we grouped the results into the categories: “lectins detecting the secreted footprints” (eight lectins) and “lectin labeling of universal and positional distinct *Hydra* cell types and associated structures” (nine lectins). From the nine lectins that did not label the footprints, four lectins (*Erythrina cristagalli* lectin, *Pisum sativum* agglutinin, *Griffonia* (*Bandeiraea*) *simplicifolia* lectin I, *Dolichos bilforus* agglutinin) detected the basal disc. Based on the labeling of the basal disc, the glycans recognized by these four lectins might play a role in adhesion, but as they were not detected in the footprints, they are likely not a major component of the adhesive material. Detailed descriptions of these results can be found in the Appendix A section: “whole-mount labeling of lectins detecting the basal disc but not the footprints” (Appendix A).

### 3.2. Lectins Detecting the Secreted Footprints

Eight lectins labeled the secreted footprints, indicating the presence of the corresponding glycans in the secreted adhesive (Table 1). In the whole-mount *Hydra* labeling, these eight lectins detected diverse structures from the overall animal surface to the nematocytes (sting cells) in the tentacles (Table 1). For example, *Wheat germ* agglutinin (WGA) (Figure 2A–D) and *Succinylated wheat germ* agglutinin (sWGA) (Figure 2E–G) detected the overall ectoderm surface (Figure 2A,B,E), the basal disc surface (Figure 2C,D), dot-like structures in the ectoderm of the tentacles, and the nematocyte capsules and operculum (Figure 2F). After Lavdowsky fixation, both lectins additionally detected the developing nematoblasts and nematocytes present in the gastric region (Figure 2G). Remarkably, the intensity and the pattern of the basal disc surface staining varied among individual *Hydras*, not relying on the fixation method (Figure 2C,D). *Datura Stramonium* lectin (DSL), *Lycopersicon esculentum* (tomato) lectin (LEL), and soybean agglutinin (SBA) detected dot-like structures and the nematocysts in the tentacles (Appendix A). DSL and SBA also reacted with the operculum, which was not stained with LEL. The basal disc surface was not detected, with the exception of an intermediate strong labeling with LEL after one hour of PFA fixation. *Ricinus communis* agglutinin (RCA) and *Ulex europaeus* agglutinin I (UEA I) labeled the overall ectoderm surface and RCA additionally labeled the basal disc surface (Appendix A). Concanavaline A (Con A) reacted strongly with the tentacle surface and all structures of the nematocytes, weakly with the overall ectoderm surface and intermediately with the basal disc surface (Appendix A).


Footprint structure


The eight lectins detecting the footprints indicated the presence of several glycans in the secreted adhesive material (Table 1). The footprint structure was the same in all lectin labelings, but the staining intensity varied. WGA led to one of the strongest signals (Figure 3), resembling previous descriptions of the *Hydra* footprint [15]. The secreted material accumulated at the basal disc cell borders, resulting in an imprint of the formerly attached basal disc cells (Figure 3A). At higher magnification, a hole in this net was occasionally present, likely because the cell at this location had not secreted its adhesive content (Figure 3B). The footprints consisted of a fibrous material resulting in an inhomogeneous, meshwork-like structure (Figure 3B). In approximately half of the observed footprints, a small-to-middle-sized area in the middle of the footprints was devoid of any secreted material (Figure 3C). On one footprint, discarded cells were observed at this location (Figure 3D). To determine if the same structures were labeled with different lectins, we performed double lectin labeling of fluorescein-conjugated WGA and the seven other lectins (Figure 3E). The difference in the staining intensity made some comparisons difficult, but the labeling overlapped WGA in all cases (Figure 3E). Four of the eight lectins (WGA, sWGA, DSL, and LEL) are known to bind to N-acetylglucosamine (GlcNAc) residues. WGA, sWGA, DSL, and LEL share similar binding preferences and bind to multimers of GlcNAc, chitobiose, and terminal GlcNAc. With the exception of DSL, these four lectins led to the strongest footprint labeling. With DSL, the intensity varied among individual footprints from weak to strong (Table 1). SBA and RCA detect N-acetylgalactosamine (GalNAc) and galactose residues. SBA led to a strong labeling, while for RCA the intensity varied in between footprints. A similar variability was observed with UEA I, a lectin reacting with α-linked fucose. Con A caused an intermediate footprint labeling and is known to bind α-linked mannose. Based on these results, we presume that the *Hydra* adhesive footprint contains GlcNAc, GalNAc, fucose, and mannose to varying degrees. 

Throughout our labeling experiments, we observed many footprints that were folded in at the edges (Figure 3A) and/or smudgy in some areas. Additionally, the basal disc surface pattern appeared variant between individual *Hydras*. We assumed that this was an artefact caused by our method to collect the footprints and *Hydras* by forcibly detaching them from the surface with a glass pipet. To test this, we let *Hydras* attach to glass slides and observed how long they stayed attached before voluntarily detaching on their own. The moment a *Hydra* detached, the footprint and corresponding *Hydra* were immediately fixed in PFA and lectin labeling with fluorescein-conjugated WGA was performed (*n* = 15). Furthermore, half of the *Hydras* were forcibly detached with a glass pipet and fixed and stained the same way (*n* = 16). The only difference we could observe between voluntarily self-detached and forcibly detached specimens was the tissue integrity of the basal disc (Figure 4). While, in voluntarily detached *Hydras,* the basal disc was always intact (15 out of 15) (Figure 4A), in forcibly detached animals the basal disc was damaged in half of the *Hydras* (8 out of 16) (Figure 4C,E). We again noted a variation in the staining intensity and appearance of the basal disc surface, but could not correlate those variations to the mode of detachment or the time the *Hydra* had stayed attached before fixation. On most basal discs, the cell borders were strongly stained, causing the characteristic net-like pattern (Figure 4A). This labeling likely stems from the adhesive material accumulating there during attachment. However, we also frequently observed a staining of the cell surfaces, likely representing freshly secreted adhesive (Figure 4E). Occasionally, footprints appeared thick and blurry, but surprisingly this was independent of their attachment time (the example in Figure 4B only attached for 11 min). Additionally, if footprints were smudgy, folded in, or incomplete (Figure 4D,F) was independent of attachment time and mode of detachment.


Localization of footprint-specific glycans within the basal disc cells 


As we were unable to visualize basal disc granules in the whole-mount samples, we performed lectin labeling of the eight footprint-specific lectins on separated (macerated) basal disc cells (Figure 5). Four secretory granule types can be distinguished in these cells: the large HSGI and HSGII, with HSGI likely representing immature HSGII, and the smaller and numerous HSGIII and HSGIV (Figure 5A) [15]. Basal disc cells are characterized by the four types of secretory granules, which are denser at the aboral end of the cells, by irregular water vacuoles, and by oriented actin filament bundles (myonemes) (Figure 5A) [15]. As expected, the four lectins WGA, sWGA, UEA I, and RCA (Figure 5B–E) reacted with numerous granules, which accumulated at the aboral end of the cells (Figure 5D,E). Based on their size and localization, they likely correspond to type III and/or IV granules. The larger HSGI and HSGII were not labeled. Surprisingly, no granular staining was detected for LEL, SBA, DSL, or Con A (Figure 5F–I). LEL and SBA reacted with vacuoles and small intracellular structures, which could not be identified (Figure 5F,G). DSL labeling of basal disc cells resulted in no or occasionally a weak labeling of vacuoles (Figure 5H). Con A reacted strongly with the cytoplasm throughout the cell, leaving only the nucleus unlabeled (Figure 5I). Overall, these results showed that the footprint material detected by WGA, sWGA, UEA I, and RCA was produced and secreted from the HSGIII and/or HSGIV granules. For the other four lectins, the origin of the detected glycans in the footprints remained unclear. 

### 3.3. Lectin Labeling of Universal and Positional Distinct Hydra Cell Types and Associated Structures

In addition to the glycans potentially involved in adhesion, the lectin screen revealed some common glycan patterns in whole-mount *Hydras*. The most prevalent stained structures were the overall ectoderm surface (14), nematocytes (12), and dot-like structures in the ectoderm (7) (Table 1). Exemplary images of frequent staining results are shown on the example of *Lens culinaris* agglutinin (LCA) and *Vicia villosa* agglutinin (VVL) in Figure 6. LCA strongly reacted with substructures of the nematocytes, with fixative-depending pattern. After PFA fixation, LCA only reacted with the base of the cnidocil (Figure 6A,B), whereas after Lavdowsky fixation, the whole capsules of the nematocytes were labeled (Figure 6C). Additionally, nematoblasts in the gastric region, as well as the surface of the tentacles, were intermediately stained and erupted tubules of nematocytes were labeled. VVL weakly labeled dot-like structures in the ectoderm of the gastric region and the tentacles (Figure 6D). These structures likely correspond to the subapical secretory granules of the ectodermal cells. Additionally, various parts of the nematocytes were labeled (Figure 6E,F). The nematocyte staining varied among fixatives, with the cnidocil only being stained after PFA fixation (Figure 6E), and not after Lavdowsky fixation (Figure 6F).

In addition to these frequent labeling patterns, some unexpected labeling results occurred in whole-mount *Hydra*. For example, PHA-L and PHA-E labeled structures associated with radial muscle fibers and GSL II detected endodermal gland cells in the hypostome of developing buds (Appendix A). Detailed descriptions of these non-adhesion related results are presented in the Appendix A section: “lectin labeling of universal and positional distinct *Hydra* cell types and associated structures”.

## 4. Discussion 

### 4.1. Glycan Distribution in Whole-Mount Hydra 

Ultrasensitive mass spectrometry has revealed that the overall *Hydra* glycome consists of heavily fucosylated N- and O-glycans [24]. As these experiments have been performed with whole *Hydra* polyps, the localization of the detected glycans has not been determined. We used a set of 23 commercially available lectins to determine glycan distribution within whole-mount *Hydras* and to distinguish glycans present in its secreted adhesive material. The lectins were selected to cover a wide range of common glycan moieties. *Hydra* is covered by a thick, layered glycocalyx composed of a high concentration of sulfated glycosaminoglycans (GAG) [25]. The glycocalyx is only well-preserved when cryo-based fixation methods (high-pressure freezing and freeze-substitution) are used [25,26]. With standard chemical fixation, such as the fixatives that were used (PFA and Lavdowsky), the glycocalyx shrinks and outer layers are lost. Nonetheless, 14 out of the tested 23 lectins labeled the overall ectoderm surface in varying degrees. Due to the technical limitations of the chemical fixation, this list is likely not exhaustive. In addition to the overall surface staining, seven lectins detected dot-like structures in the ectoderm. Based on their position and size, they likely corresponded to subapical secretory granules of the ectodermal epithelial cells, which are secreting the glyocalyx’s components [25]. Mostly, the labeling appeared stronger at the tentacles compared to the gastric region. If this was an artefact due to the poor conservation of the glycocalyx or if there is a positional difference in the glycocalyx’s composition remains unknown. 

### 4.2. Hydra Footprints Are Built up by a Fibrillar Material

In other temporarily adhering animals, such as sea stars, the amount of secreted adhesive material varies depending on the surface composition [27] and the strength and duration of attachment [28]. In *Hydra*, no correlation between the thickness of the footprints and attachment time could be determined. Furthermore, sea star footprints are formed by a thin homogenous film covering the surface and a thick meshwork on top of it [8,27]. In contrast to this, our labeling revealed that *Hydra* footprints were formed by a fibrillar, dense meshwork. Basal disc cell borders could be distinguished by an accumulation of adhesive material, resembling previous descriptions [15]. Additionally, we occasionally observed holes in the net, probably resulting from the non-secretion of the basal disc cell at this location. In half of all footprints, the middle area was devoid of any secreted material. This could potentially indicate that *Hydra* is able to create a vacuum under its basal disc to increase attachment strength. Alternatively, this could also result from the aboral pore located in the middle of the basal disc [29] and/or old epidermal cells, which are supposed to be discarded at this location. In one case, we observed discarded cells in the middle of the footprints, supporting the latter explanation. 

The process of voluntary detachment from a secreted adhesive is still unclear. In the marine flatworm *Macrostomum lignano*, it has been proposed that a negatively charged molecule is secreted and outcompetes the binding of the positively charged adhesive to the glycocalyx [13]. In sea stars and sea urchins, an enzymatic detachment through proteinases has been postulated [30,31]. In contrast to these animals, *Hydra* lacks a dedicated de-adhesive gland. In *Hydra* the position and orientation of myonemes in its basal disc might allow for a mechanical detachment. Moreover, video analyzes of detaching individuals support the theory that muscular contractions in the basal disc are involved in the detachment process [15]. In our study, many footprints appeared to be smeared and folded in at the rim, regardless of whether the polyps were detached by force or detached voluntarily on their own. This observation is in line with the theory of a mechanical detachment. 

### 4.3. Glycans Detected in the Secreted Footprint

We identified eight lectins that reacted with the secreted *Hydra* footprint. Surprisingly, only four (WGA, sWGA, UEA I, and RCA) equally detected granules in the basal disc cells, the subcellular structures in which the adhesive is stored until secretion [15]. All four lectins labeled small numerous granules that likely correspond to type III or IV. Both granule types have been described to be PAS-positive, highlighting that they are rich in glycans [15]. That the other four lectins (DSL, LEL, SBA, and ConA) did not react with any secretory granules could be an artefact caused by a limited accessibility of the glycans in the densely packed granules, as has previously been observed in sea star adhesive granules [7]. 

The fact that glycans are prevalent in temporary adhesives indicate an essential role in the adhesion process, but their function is still speculative. It has been proposed that cohesive strength is achieved through glycan–protein interactions, involving glycoproteins and proteins with glycan-binding functional domains [13]. Glycosylation could also enhance protein-binding ability and make proteins more resistant to degradation [6]. There is a high variability of glycans in the adhesive material found in between species [6,10,11,32,33]. Even in animals of the same phylum, the adhesive glycan composition is variant. For example, in the sea urchin *Paracentrotus lividus*, five lectins (GSL II, WGA, STL, LEL, and SBA) label the adhesive material [5], but in three other sea urchin species these lectins lead to different results [6]. In the sea star *Asterias rubens*, four lectins detect the footprints (DBA, WGA, RCA, and Con A), while in *Asterina gibbosa* this was only true for one lectin (Con A) [7,8]. This variability might be caused by an adaptation of the species to their respective environment, but further research is required to unravel the cause and functional relevance of this inconsistency in the glycan composition. 

The glycans detected in the secreted adhesive are often part of the glycosylated proteins [5,6,7,12,13,34]. However, their function during attachment is mostly unknown. In *M. lignano*, the function of a glycosylated adhesive protein has been determined [13]. The adhesive protein is associated with GalNac residues and can be detected by the lectin PNA [9,13]. The glycoprotein binds to the surface during attachment and, upon functional knock-down, the animals are unable to attach themselves [13]. In *Hydra*, eight lectins reacted with the footprints left on the substrate, indicating that the *Hydra* adhesive contains GlcNAc, GalNAc, fucose, and mannose to varying degrees. It is unknown if these glycans are part of glycosylated proteins. Nonetheless, the presence of the enzyme glycosyl hydrolase AbfB [21] and three subunits of a Dolichyl-diphospho-oligosaccharide-protein glycosyltransferase [20] in the basal disc cells indicate that at least some glycans might be part of glycosylated proteins.

### 4.4. Lectins as Markers for Hydra Nematocytes

We found that 17 out of 23 lectins labeled whole-mount *Hydra* in a distinct pattern. Notably, 12 lectins reacted with fully differentiated nematocytes in the tentacles (Table 1). In *Hydra*, four different types of nematocytes can be distinguished: the holotrichous and the atrichous isorhizas (spineless), the desmonemes (small and with a tightly coiled tubule) and the stenoteles (large and with a prominent stiletto apparatus at the base of their tubules) [35]. In the tentacles, the mature nematocytes are incorporated into large battery cells, containing all the different types [36]. We observed no difference in the lectin labeling for the four types, except that sometimes the labeling intensity varied slightly. These results indicated that the glycan composition was similar among all four nematocyte types.

The nematocyte capsules (nematocysts) consist of an extracellular matrix-like composition of proteins and GAG, and protein–carbohydrate interactions mediate their capsule assembly [35]. Additional to structural proteins, such as minicollagens, C-type lectin NOWA, and spinalin [37], nematocysts are rich in chondroitin, which is a sulfated GAG, composed of a chain of alternating sugars (GalNAc and glucuronic acid) [38]. The chondroitin is present in form of proteoglycans [38] and GAG biosynthesis inhibition, using a β-D-Xyloside treatment, results in the complete absence of mature nematocysts in the tentacles [39]. This indicates that the GAG plays a crucial role in the capsule assembly and might serve as a scaffold for the structural proteins [39]. Our results confirm that nematocytes contain a high amount of glycans and indicate the presence of GluNAc, GalNAc, and mannose residues in the capsules, the operculum, and the cnidocil. Anti-chondroitin antibodies mainly react with differentiating nematoblasts, whereas, in mature nematocytes, only the operculum is stained [39]. The capsules’ walls harden during maturation, which might limit the antibodies’ access [39]. Accordingly, we could not observe any labeling of the nematocysts’ tubules in intact nematocytes, but several lectins labeled erupted nematocysts’ tubules. Furthermore, the lectin STL caused a blurry labeling surrounding erupted tubules, which might indicate that the content of the capsules also contained glycans. However, erupted nematocysts were not observed in all samples; therefore, our results might not be exhaustive. Additionally, the fixation method influenced the lectin labeling outcome. Mature nematocytes in the tentacles were labeled after both fixations, but developing nematoblasts were only stained after Lavdowsky fixation. 

### 4.5. Biomimetic Approaches and Their Limitations

Adhesives that perform under wet conditions or even underwater would have broad applications in the engineering and medical fields. Natural, aquatic adhesives might serve as a source for bio-inspired synthetic counterparts [40]. Thus far, biomimetic approaches mainly focused on adhesives produced by permanent adhering animals, like mussels [41]. In recent years, the adhesives produced by temporarily adhering animals have gained increasing attention. In contrast to permanent adhesion, temporarily adhering animals can repeatedly detach and reattach [4]. The involved adhesive proteins are not conserved among phyla, but share reoccurring characteristics, such as a biased amino acid distribution, repetitive regions, and prevalent protein domains [2]. For example, the cohesive proteins of sea stars, sea urchins, limpets, and flatworms contain calcium-binding epidermal growth factor (EGF)-like domains, galactose-binding lectin domains, discoidin domains (also known as F5/8 type C domains), von Willebrand Factor type D domains, and trypsin inhibitor-like cysteine rich domains [12,13,30,42]. Two fragments of the sea star cohesive protein that comprise these domains have been recombinantly produced in bacteria [43,44]. These recombinant proteins not only self-assemble and adsorb on various surfaces, they also show no cytotoxic effects on cell cultures [43]. These results are highly promising and show the potential of recombinantly produced adhesive proteins for biomedical applications. Nevertheless, the approach has its limitations, as recombinant production via bacteria is restricted to single proteins and fails to reproduce any post-translational modifications of the proteins. The natural sea star adhesive consists of a set of 16 proteins [28], of which many are glycosylated [7]. The recombinant proteins, therefore, only represent a fraction of the natural adhesive. To replicate the adhesive strength achieved in the natural system, the protein interactions and the role of the prevalent glycans need to be investigated. However, tools to test gene and protein function in sea stars are not available. In *Hydra*, the needed molecular tools, such as gene knock-down and knock-out, are well established. Previous findings show that the *Hydra* adhesive contains proteins with glycan-binding domains [21]. Here, we identified the glycans N-acetylglucosamine, N-acetylgalactosamine, fucose, and mannose in the adhesive, which might be relevant to the proposed glycan–protein interactions. Our findings now lay the basis for further functional investigations on glycan and protein function. 

## 5. Conclusions

Bio-inspired adhesives present themselves as a high potential substitute to the currently used synthetic adhesives. The unraveling of the molecular composition of bioadhesives is crucial to provide models for bio-inspired technologies. *Hydra* constitutes a standard model in stem cell biology and tissue regeneration, but few studies have addressed its underwater attachment ability. This project aimed to identify the glycans present in the *Hydra* secreted adhesive material, complementing previous transcriptomic and proteomic work. Our results indicate the presence of N-acetylglucosamine, N-acetylgalactosamine, fucose, and mannose in the secreted adhesive material. Furthermore, we observed a meshwork-like substructure in the footprints that implies that the adhesive is mainly formed by fibers. Additionally, we showed that commercially available lectins can be used as markers for several *Hydra* cell types and structures, such as nematocytes, endodermal gland cells, and cell membranes.

## Figures and Tables

**Figure 1 biomimetics-07-00166-f001:**
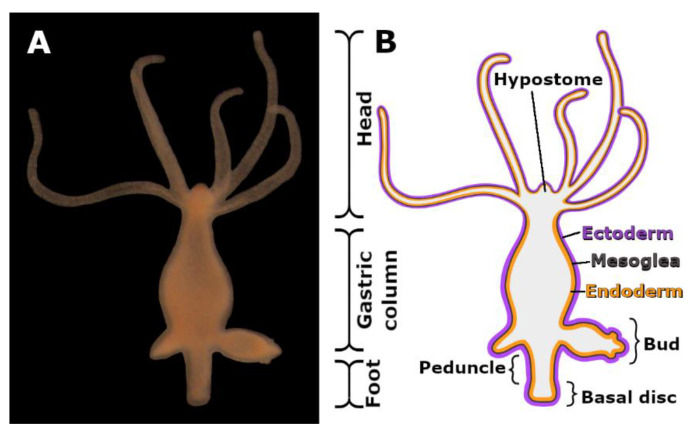
(**A**) Picture and (**B**) morphological scheme of an adult asexually reproducing *Hydra* polyp.

**Figure 2 biomimetics-07-00166-f002:**
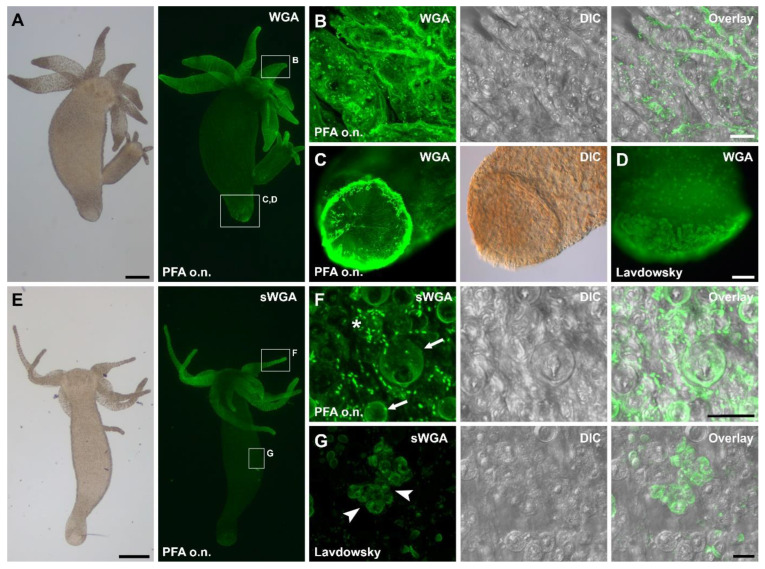
Lectin labeling of *Hydra* whole-mounts with (**A**–**D**) WGA and (**E**–**G**) sWGA. (**A**) WGA and (**E**) sWGA labeling of whole-mount individuals. (**B**,**F**) Overall ectoderm surface, ellipsoid structures, nematocytes capsules, and operculum were labeled with both lectins. Basal disc surface staining for (**C**) overnight PFA and (**D**) Lavdowsky fixatives. Note that the intensity and the pattern of the disc surface staining varied among individual *Hydras*, without relying on the fixation method. (**G**) After Lavdowsky fixation, developing nematoblasts and nematocytes were labeled in the body column for both lectins (sWGA is shown in the figure). Arrows highlight nematocytes and arrowheads point towards developing nematoblasts. The asterisk indicates ellipsoid structures. The fixation method used is indicated in the images. Scale bars: (**A**,**E**) 500 µm; (**C**,**D**) 100 µm; (**B**,**F**,**G**) 20 µm.

**Figure 3 biomimetics-07-00166-f003:**
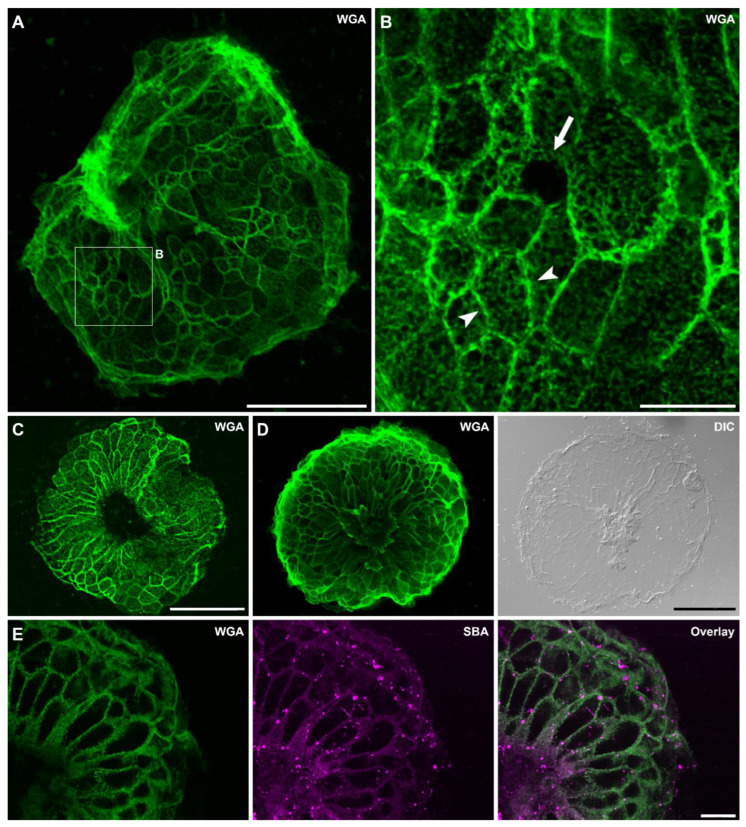
Lectin labeling of *Hydra* footprints with WGA (**A**–**D**) and SBA/WGA double labeling (**E**). (**A**) Footprint overview. (**B**) Footprint structure at a higher magnification; note the meshwork-like appearance. Arrowheads highlight the imprint of the basal disc cell borders and the arrow indicates a hole in the footprint. (**C**) Footprint with an empty area in the middle and (**D**) discarded cells at the same position. **(E)** Footprint detail of SBA and WGA double staining, note that the staining overlaps. Scale bars: (**A**,**C**,**D**) 100 µm; (**B**,**E**) 20 µm.

**Figure 4 biomimetics-07-00166-f004:**
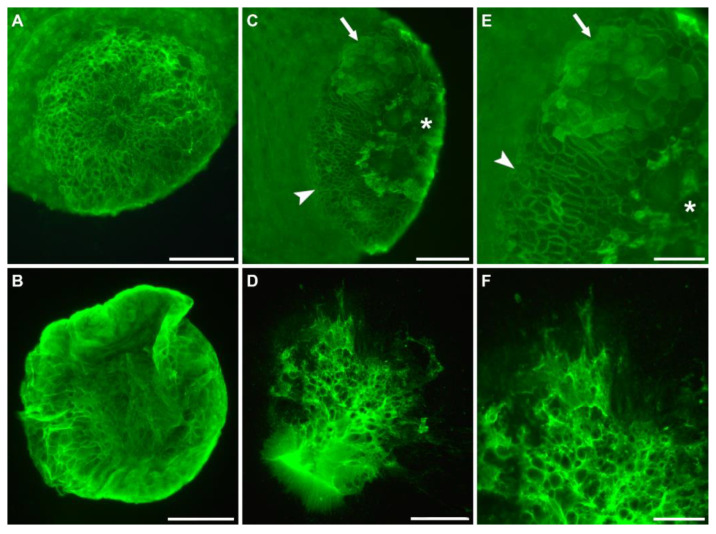
Representative lectin labeling of *Hydra* basal disc and the corresponding footprint with WGA. (**A**) Basal disc of a voluntarily detached *Hydra* and (**B**) corresponding footprint after 11 min attachment. (**C**,**E**) Basal disc of a forcibly detached animal and (**D**,**F**) corresponding footprint after 78 min attachment. Arrowheads indicate the adhesive material accumulated at the basal disc cell borders and arrows point to the adhesive material on the surface of the cells. Asterisks indicate damaged tissue. Scale bars: (**A**–**D**) 100 µm; (**E**,**F**) 50 µm.

**Figure 5 biomimetics-07-00166-f005:**
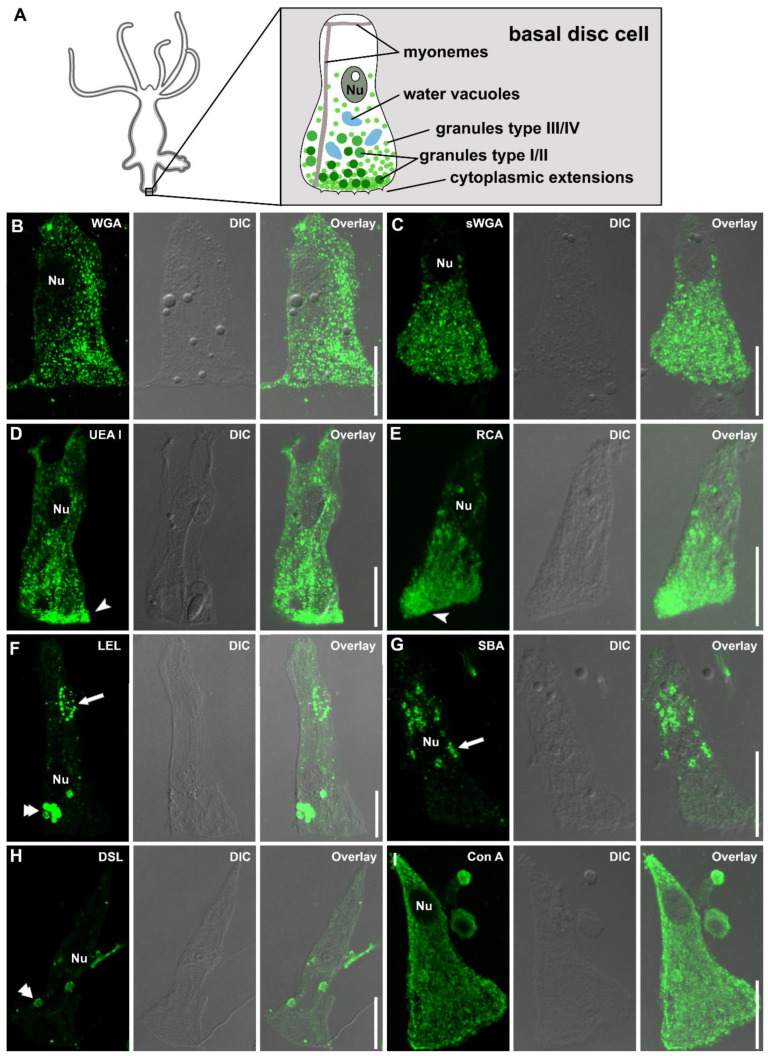
(**A**) Morphological scheme and (**B**–**I**) lectin labeling of macerated *Hydra* basal disc cells. Lectin labeling with (**B**) WGA, (**C**) sWGA, (**D**) UEA I, (**E**) RCA, (**F**) LEL, (**G**) SBA, (**H**) DSL, and (**I**) Con A. Arrow heads indicate denser concentrations of granules, double arrows point to labeled vacuoles and arrows to the dot-like structures. Scale bars: 20 µm.

**Figure 6 biomimetics-07-00166-f006:**
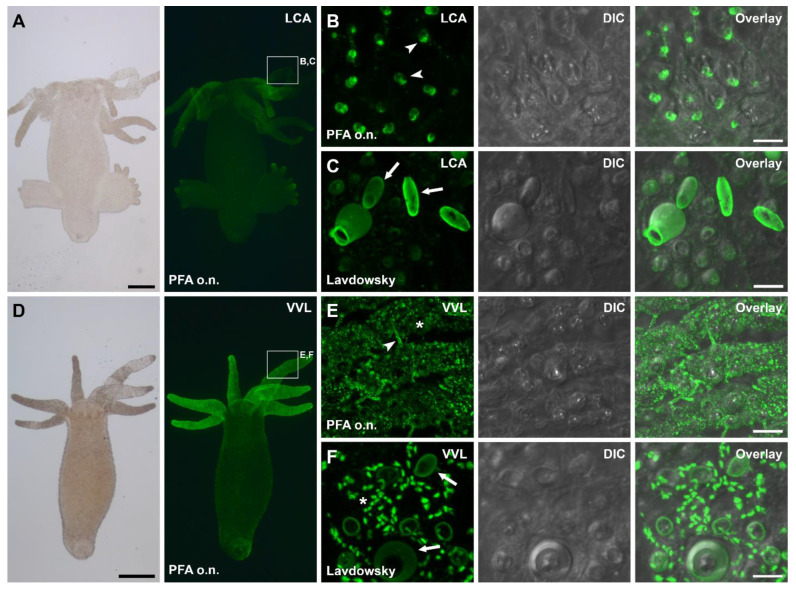
Lectin labeling of *Hydra* whole-mounts with (**A**–**C**) LCA and (**D**–**F**) VVL. (**A**–C) LCA labeling of a (**A**) whole-mount individual. Detailed view of stained structures in the tentacles, showing (**B**) cnidocil base staining and (**C**) nematocyst staining, fixed with PFA overnight or Lavdwosky for 4 h, respectively. (**D**–**F**) VVL labeling of a (**D**) whole-mount individual. Detailed view of stained structures in the tentacles, showing (**E**) cnidocil, nematocytes and dot-like structures and (**F**) nematocytes and dot-like structures, fixed with PFA overnight or Lavdwosky for 4 h, respectively. The fixation method used is indicated in the images. Arrowheads highlight the cnidocil base and the cnidocil, arrows indicate the nematocytes capsules and asterisks the dot-like structures. Scale bars: (**A**,**D**) 500 µm; (**B**,**C**,**E**,**F**) 10 µm.

## Data Availability

Not applicable.

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
