# Peer review of "The Involvement of Cell-Type-Specific Glycans in Hydra Temporary Adhesion Revealed by a Lectin Screen"

_biomimetics, 2022, doi:10.3390/biomimetics7040166_

Round 1

Reviewer 1 Report

The manuscript entitled ‘A lectin screen reveals cell type-specific glycan patterns in Hydra and indicates the involvement of glycans in its temporary adhesion’ by Seabra et al aimed to identify the glycans present in the adhesive material secreted by the solitary polyp Hydra. The results indicated the presence of N-acetylglucosamine, N-acetylgalactosamine, fucose, and mannose in the Hydra adhesive. The authors observed a meshwork-like substructure in the footprint and also showed that commercially available lectins can be used as markers for several Hydra cell types and structures, like nematocytes, endodermal gland cells and cell membranes. This is an interesting study and the results can deepen our understanding of the temporary adhesion and detachment mechanism in aquatic ecosystems, especially the involvement of glycans in underwater adhesion. I recommend it for publication after minor revision.

Minor comments

1. The title ‘The involvement of cell type-specific glycans in Hydra temporary adhesion revealed by lectin screen’ may be a concise for this manuscript.

2. Introduction section. The research progress of lectin's function in aquatic organism adhesion should be introduced.

3. Discussion section. Lines 369-376. The author described that due to the fixatives used (PFA or Lavdowsky) the glycocalyx was shrunk and poorly conserved in samples. If changing these fixatives can solve the problems of shrinkage and preservation, it can solve the problem about glycocalyx raised at the end of this paragraph. Authors should not leave the problems of experimental techniques to readers.

Author Response

We thank the two reviewers for the thoughtful comments and suggestions. We have changed the manuscript according to their suggestions and believe that the quality and clarity of the paper improved.

Reviewer 1:

The manuscript entitled ‘A lectin screen reveals cell type-specific glycan patterns in Hydra and indicates the involvement of glycans in its temporary adhesion’ by Seabra et al aimed to identify the glycans present in the adhesive material secreted by the solitary polyp Hydra. The results indicated the presence of N-acetylglucosamine, N-acetylgalactosamine, fucose, and mannose in the Hydra adhesive. The authors observed a meshwork-like substructure in the footprint and also showed that commercially available lectins can be used as markers for several Hydra cell types and structures, like nematocytes, endodermal gland cells and cell membranes. This is an interesting study and the results can deepen our understanding of the temporary adhesion and detachment mechanism in aquatic ecosystems, especially the involvement of glycans in underwater adhesion. I recommend it for publication after minor revision.

Minor comments

  1. The title ‘The involvement of cell type-specific glycans in Hydra temporary adhesion revealed by lectin screen’ may be a concise for this manuscript.

We thank the reviewer for this suggestion and have changed the title accordingly.  

  1. Introduction section. The research progress of lectin's function in aquatic organism adhesion should be introduced.

As also suggested by Reviewer 2, we have shortened the general introduction of the model organism Hydra and added a short section on the research on the composition of aquatic adhesives. It now reads:

"The ability of organisms to attach to surfaces is entitled biological adhesion and is present in various organisms. Bioadhesion processes are diverse and complex and play a crucial role in organism survival and basic functions [1]. In glue-based adhesive systems, the attachment is mediated by the secretion of an adhesive material and can either be temporary or permanent [2]. The biochemical composition of the secreted adhesive material varies among organisms and is difficult to characterize [3]. Common ground states that in temporarily adhering animals, like Hydra, it is mainly constituted by proteins and carbohydrates [2,4]. Studies on aquatic temporary adhesives predominately focus on the identification of proteins, but especially in temporary adhesion, carbohydrates are abundant in the secreted material [2]. Adhesion-related glycans have been mostly detected through histological stains like Alcain blue and lectin-binding assays [2]. Using lectin based methods, glycans have been consistently detected in the adhesive of non-permanently adhering animals like sea urchins [5,6], sea stars [7,8], flatworms [9-11], and limpets [12]. Moreover, aquatic adhesive proteins are often highly glycosylated [5-7,13,14]. This post-translational modification significantly changes proteins characteristics and has to be taken into account in any biomimetic approach."

  1. Discussion section. Lines 369-376. The author described that due to the fixatives used (PFA or Lavdowsky) the glycocalyx was shrunk and poorly conserved in samples. If changing these fixatives can solve the problems of shrinkage and preservation, it can solve the problem about glycocalyx raised at the end of this paragraph. Authors should not leave the problems of experimental techniques to readers.

We are sorry that this paragraph was confusing for readers and we changed the text according to the suggestions. Chemical fixation is not sufficient to preserve all the layers of the glycocalyx in Hydra, this can only be achieved through high-pressure freezing methods. To clarify, we changed the text to the following:

"The glycocalyx is only well preserved when cryo-based fixation methods (high-pressure freezing and freeze-substitution) are used [25,26]. With standard chemical fixation, as the fixatives used (PFA and Lavdowsky), the glycocalyx shrinks and outer layers are lost. Nonetheless, 14 out of the tested 23 lectins labeled the overall ectoderm surface in varying degrees. Due to the technical limitations of the chemical fixation, this list is likely not exhaustive."

Reviewer 2 Report

This paper has several interesting findings, but there is a great deal of additional descriptive information that is not relevant to biomimetics and bioadhesion. The findings that are of interest are that the footprint has a mesh-like structure (Figure 5), and that the larger secretory granules (HSGI and HSGII) were not labeled whereas HSGIII and/or HSGIV were labeled by WGA, sWGA UEA I and RCA, which also labeled the footprint material. I would like to see more work building on these interesting findings.

The rest of the information seems extraneous. From the point of view of biomimetics and bioadhesion, any glycans present in the upper body, gastric regions, buds, tentacles, nematocysts or other such areas is not relevant. I think a general overview of glycan staining throughout hydra will not be of interest to readers of this journal. There are obviously many different glycosylated proteins in an animal.

Another issue is highlighted by the authors when they discuss the variability among species in glycans present in glues. The fact that there is so much variation, and we don’t know whether or not it matters, weakens the paper. 

Also note that the 2nd paragraph of the Introduction provides too much general background information on hydra. It should stay focused on biomimetic aspects and adhesion. Fig 1 doesn’t seem relevant to the paper.

Overall, there is a core of interesting work here that should be followed up on, but right now it is too limited and preliminary.

Author Response

We want to thank the reviewer for the critical assessment of our manuscript. We agree that the detailed description of the lectin screen impaired the readability and this part is not relevant for researchers in the field of adhesion. We initially included these data, as they are of interest for researchers in the large cnidarian community. We now shifted the focus towards the results relevant for adhesion and moved the description of more general glycan pattern into the Supplementary materials.

As suggested, we strongly shortened the general background information on Hydra in the introduction and put more focus on the composition of aquatic adhesives. We decided to keep (a shortened) version of Fig. 1, as most readers will be unfamiliar with the model organism.

We have changed parts of the discussion to highlight the importance of a better characterization of glycans in aquatic species. Additionally, we included a section on biomimetic approaches and limitations, it now reads:

"Adhesives that perform under wet conditions or even underwater would have broad applications in the engineering and medial fields. Natural, aquatic adhesives therefore might serve as a source for bio-inspired synthetic counterparts [40]. So far, biomimetic approaches mainly focused on adhesives produced by permanent adhering animals, like mussels [41]. In recent years, the adhesives produced by temporarily adhering animals have gained increasing attention. In contrast to permanent adhesion, temporarily adhering animals can repeatedly detach and reattach [4]. Involved adhesive proteins are not conserved among phyla, but they share reoccurring characteristics, like biased amino acid distribution, repetitive regions, and prevalent proteins domains [2]. For example, the cohesive proteins of sea stars, sea urchins, limpets, and flatworms contain calcium-binding epidermal growth factor (EGF)-like domains, galactose-binding lectin domains, discoidin domains (also known as F5/8 type C domains), von Willebrand Factor type D domains, and trypsin inhibitor-like cysteine rich domains [12,13,30,42]. Two fragments of the sea star cohesive protein (Sfp1) that comprise these domains have been recombinantly produced in bacteria [43,44]. These recombinant proteins not only self-assemble and adsorb on various surfaces, they also show no cytotoxic effects on cell cultures [43]. These results are highly promising and show the potential of recombinantly produced adhesive proteins for biomedical applications. Nevertheless, the approach has its limitations, as recombinant production via bacteria is restricted to single proteins and fails to reproduce any post-translational modifications of the proteins. The natural sea star adhesive consists of a set of 16 proteins [28], of which many are glycosylated [7]. The recombinant proteins therefore only represent a fraction of the natural adhesive. To replicate the adhesive strength achieved in the natural system, the protein interactions and the role of the prevalent glycans need to be investigated. However, tools to test gene and protein function in sea stars are not available. In Hydra, the needed molecular tools like gene knock-down and knock-out are well established. Previous findings show that the Hydra adhesive contains proteins with glycan-binding domains [21]. Here, we identified the glycans N-acetylglucosamine, N-acetylgalactosamine, fucose, and mannose in the adhesive, which might be relevant for the proposed glycan-protein interactions. Our findings now lay the basis for further functional investigations on glycan and protein function."

Round 2

Reviewer 2 Report

There are a few interesting findings, but the authors need to build upon them to make a complete paper. As it stands the main findings have some interest but more experimental work needs to be done.